# Rare emm6.10 *Streptococcus pyogenes* Causing an Unusual Invasive Infection in a Child: Clinical and Genomic Insights

**DOI:** 10.3390/microorganisms13112475

**Published:** 2025-10-29

**Authors:** Laurent Blairon, Marie Tré-Hardy, Veerle Matheeussen, Sien De Koster, Marie Cassart, Sarah Heenen, Andrea Nebbioso, Nancy Vitali

**Affiliations:** 1Department of Laboratory Medicine, Iris Hospitals South, 1050 Brussels, Belgium; lblairon@his-izz.be; 2Department of Pharmacy, Namur Research Institute for Life Sciences, University of Namur, 5000 Namur, Belgium; 3Faculty of Medicine, Université libre de Bruxelles, 1070 Brussels, Belgium; 4National Reference Centre for Invasive β-Hemolytic Streptococci Non-Group B, Antwerp University Hospital, 2650 Edegem, Belgium; 5Department of Medical Imaging, Iris Hospitals South, 1050 Brussels, Belgium; 6Department of Intensive Care Medicine, Iris Hospitals South, 1050 Brussels, Belgium; 7Department of Pediatrics, Iris Hospitals South, 1050 Brussels, Belgium

**Keywords:** Group A *Streptococcus*, *emm* typing, invasive infection, periostitis, children

## Abstract

Invasive group A streptococcal (iGAS) infections are increasingly recognized as a global public health concern, with a notable resurgence observed among pediatric populations in high-income countries following the relaxation of COVID-19-related restrictions. While the most commonly implicated emm types in invasive disease are emm1 and emm3, the global distribution of *Streptococcus pyogenes* strains is highly diverse, posing challenges for surveillance and vaccine development. We describe a 3-year-old boy with a femoral subperiosteal abscess, a rare clinical manifestation of iGAS, caused by an emm6.10 *S. pyogenes* strain. The diagnosis was confirmed by positive blood cultures and magnetic resonance imaging. Antibiotic therapy included intravenous ceftriaxone followed by oral amoxicillin, and then prolonged oral clindamycin was introduced due to the deep-seated nature of the infection. Molecular typing was performed by the national reference center as part of routine surveillance of invasive strains. This case emphasizes the importance of recognizing atypical clinical presentations of iGAS in children and the crucial role of strain typing in epidemiological monitoring. It also illustrates how the remarkable emm-type diversity of *S. pyogenes* remains a major obstacle to effective vaccine design, despite ongoing efforts with multivalent M-protein-based candidates and alternative strategies targeting conserved antigens. Enhanced global surveillance and inclusive vaccine design are urgently needed to address the full spectrum of circulating GAS strains.

## 1. Introduction

Invasive group A streptococcal (iGAS) infections are increasingly reported worldwide, particularly among children [1,2]. Recent pediatric outbreaks in Europe have led to public health alerts, and the World Health Organization has emphasized the need for enhanced surveillance [3].

Beyond these recent alerts in high-income countries, *Streptococcus pyogenes* (Group A *Streptococcus*, GAS) also causes a disproportionate burden of disease in low- and middle-income countries, especially among children under five years of age [4,5]. In these settings, GAS is not only responsible for acute infections but also for significant long-term morbidity and mortality due to post-infectious immune-mediated diseases such as acute rheumatic fever and rheumatic heart disease. According to Carapetis et al., the global burden of GAS-related diseases exceeds 500,000 deaths annually, underscoring the urgent need for improved surveillance and prevention strategies worldwide [6,7].

Although *S. pyogenes* most commonly causes benign conditions such as pharyngitis and superficial skin infections, its invasive forms can result in severe and life-threatening diseases, including necrotizing fasciitis, toxic shock syndrome, and bacteremia [8]. In recent years, a marked increase in pediatric iGAS incidence has been observed, particularly following the relaxation of COVID-19-related public health restrictions, with rates reaching 6 cases per 100,000 children [9]. This trend was recently confirmed in a multi-country review describing the 2022–2023 resurgence of iGAS in Europe and other high-income settings, highlighting the roles of post-pandemic waned immunity and viral co-infections [1,2].

Here, we present the case of a 3-year-old boy with an invasive group A streptococcal infection manifesting as a subperiosteal abscess of the femur, an unusual clinical presentation, caused by an emm6-type strain. Osteoarticular forms of iGAS are rarely described in children, and reports of subperiosteal abscesses are exceptionally uncommon in the literature [10]. Through this case, we aim to raise awareness of atypical manifestations of pediatric iGAS and underscore the importance of strain typing and genomic characterization in elucidating their clinical and epidemiological significance.

## 2. Clinical Case

A 3-year-old boy presents to the pediatric emergency department of a public hospital in Brussels, Belgium, with a fever up to 40.5 °C for 3 days. No recent travel was reported. On the first day of symptoms, he was seen by the attending physician, who diagnosed acute otitis media and tonsillitis. Symptomatic treatment was recommended. On the third day of illness, the patient complained of pain in the lower right limb, accompanied by lameness, prompting presentation to the emergency department. The clinical examination revealed an altered general condition with pain on mobilization of the lower right limb and an inability to bend the right leg at the hip. No visible skin signs (erythema, edema, warmth). Laboratory tests showed a significant inflammatory response with CRP at 300 mg/L (normal: <5 mg/L), leukocyte count at 7060/μL (normal: 4900–12,900/μL), and neutrophils at 88%. A bacteriological workup was performed (blood, urine, and stool cultures), and the child was hospitalized and started on empirical intravenous antibiotic therapy with ceftriaxone (100 mg/kg/day). Blood cultures were positive for *S. pyogenes*, identified by MALDI-TOF mass spectrometry. According to EUCAST 2022 interpretation, the strain was sensitive to penicillin, erythromycin, clindamycin, tetracycline, glycopeptides, chloramphenicol, and co-trimoxazole, and resistant to quinolones. Typing of the *emm* gene was performed by the Belgian Reference Center for invasive β-hemolytic streptococci non-group B at the Antwerp University Hospital according to the CDC *emm* typing protocol [11]. To further characterize the strain, whole genome sequencing was performed using the Illumina MiSeq platform (Illumina, San Diego, CA, USA). Genomic DNA was extracted using a QIASymphony SP/AS instrument with the Complex800_OBL_V4_DSP protocol and the virus/pathogen midi kit (Qiagen, Hilden, Germany). The short read sequencing library was prepared with the Nextera XT sample preparation kit (Illumina, San Diego, CA, USA) followed by 2 × 250 bp paired end sequencing. The sequencing data was deposited in NCBI under BioProject ID PRJNA1320515. Data analysis was performed using the cloud-based platform 1928diagnostics and ABRicate v1.0.1 with the VFDB and PlasmidFinder databases [12,13,14].

Imaging showed a very thin fluid layer in the right subquadriceps bursa on knee ultrasound. On day 5 of hospitalization, given the good clinical and biological evolution (reduced lameness and CRP decreased to 47 mg/L), oral amoxicillin for 7 days (100 mg/kg/day in 4 doses) was prescribed, and outpatient follow-up was arranged. Radiological evaluation was completed with an MRI of the knee, showing a subperiosteal collection in the distal third of the posterior part of the right femur (Figure 1).

The collection could not be aspirated because of its deep location and anatomical inaccessibility. Clindamycin (40 mg/kg/day in 3 doses) was initiated for a total duration of 7 weeks. At one-month follow-up, the outcome was favorable, with complete resolution of the mobility restriction and control ultrasound showing a moderate decrease in the abscess size (Figure 2).

## 3. Discussion

GAS is a bacterium that causes a wide range of infections, often benign but sometimes very serious or even fatal when it becomes invasive. According to the CDC, a GAS infection is classified as invasive when the bacterium is isolated from a normally sterile biological site such as blood, joint fluid, cerebrospinal fluid, pleural fluid, or deep tissue, or from a usually non-sterile site in a patient with toxic shock syndrome. iGAS can occur in children or adults regardless of comorbidities, although concomitant viral infections (influenza, COVID-19), immunosuppression, neoplasms, alcoholism, diabetes, and skin barrier impairment all increase morbidity and mortality.

The invasive nature of a GAS infection is mainly due to a surface antigen, M protein, encoded by the *emm* gene. This protein is involved in colonization, resistance to phagocytosis, and invasion of sterile anatomical sites. The bacterium’s pathogenicity can be increased by toxin production, as seen in *Staphylococcus aureus* [8]. Over 275 distinct emm types have now been identified, according to recent CDC data [15]. A recent global review has highlighted the extraordinary strain diversity of *S. pyogenes*, particularly in low- and middle-income countries, where the distribution of emm types differs markedly from that observed in high-income regions. This variability has major implications for surveillance and vaccine development, as some hyperinvasive emm types remain rare and might not be adequately covered by the candidate multivalent M-protein vaccines currently under development [4]. This diversity also complicates the interpretation of rare emm types such as emm6 in invasive disease. Although emm6 is included in some multivalent M-protein vaccine candidates currently under evaluation, the overall diversity of circulating emm types underscores the challenge of achieving broad and universal vaccine coverage. A recent longitudinal study from The Gambia further illustrated the limitations of extrapolating emm-type-based surveillance from high-income countries. The authors identified a broad diversity of GAS strains, with very limited overlap with the most common types found in Europe, reinforcing concerns about the global applicability of candidate M-protein-based vaccines still under development and not yet available for clinical use [5]. In addition to geographical variability, invasiveness among emm types remains an open question. Some authors describe emm1 and emm3 as the most invasive, but several European teams have found that many emm types are involved in invasive infections [16,17,18]. This has recently been addressed in a global study analyzing over 59,000 GAS isolates, which showed that while emm1 is the most frequently found type in invasive infections in high-income countries, it ranked only 39th in terms of intrinsic invasiveness. Several other emm types, including emm33, emm82, emm75, and emm92, were found to be intrinsically more invasive, with significantly higher odds ratios compared to emm1 [19].

The strain isolated in our patient was sent to the national reference center (NRC) as recommended. NRCs for invasive β-haemolytic streptococci play a critical role in epidemiological surveillance, strain typing, and antimicrobial resistance monitoring. Their work is essential for detecting emergent lineages, understanding shifts in emm type distribution, and guiding public health responses to outbreaks of invasive streptococcal disease. In this case, the isolate was identified as emm6.10, a relatively uncommon emm type in invasive disease [20]. The frequency of the emm6 type in Europe varies by author, ranging from less than 6% to more than 21% [17,18,21]. In line with the phenotypic antibiotic susceptibility testing, no acquired antibiotic resistance genes were detected. However, a *parC* mutation (Ser79Ala) was identified, consistent with the observed quinolone resistance and previously described in *S. pyogenes* emm6 [22,23]. Genes encoding streptococcal pyrogenic exotoxins (*smeZ*, *speA*, *speC*, *speG*, *speH*, *speI*, and *speK*) were detected. *Streptocococus pyogenes* strains also produce a wide variety of extracellular virulence factors that are secreted or released from the bacterial surface [23]. These virulence factors, also identified in the emm6.10 isolate, are involved in adherence, exoenzyme production, heme uptake, immune evasion and modulation, and stress survival (Table 1). Plasmids were absent.

Specific superantigens are often associated with particular emm types commonly linked to invasive infections. For example, emm1 isolates typically harbor *speA*, *speG* and *speJ*, emm3 isolates often carry *speA*, *ssa*, *speG* and *speK*, while emm12 isolates usually contain *speC*, *speG*, *speH* and *speI* [24,25]. Compared to the well-characterized invasive emm1 and emm3 lineages, our emm6.10 isolate exhibited a broad and partially overlapping virulence gene repertoire [26]. Notably, the presence of the *speA* superantigen gene, classically associated with both emm1 and emm3, suggests acquisition or maintenance of key virulence determinants typically linked to severe iGAS phenotypes. Beyond *speA*, our strain also carried multiple additional superantigen genes (*speC*, *speG*, *speK*, *speH*, *smeZ*) and cytotoxin genes (*slo*, *sagA*), as well as capsule synthesis (*hasABC*) and immune evasion factors (*ideS*, *endoS*). This combination of toxins and immune modulators may contribute to its invasive potential, suggesting that emm6.10 can harbor virulence features comparable to those of more common invasive emm types [26,27,28].

Although invasive GAS infections were traditionally considered stable in many regions, surveillance data from countries such as the UK had already documented a rising trend prior to the COVID-19 pandemic [27]. This was followed by a transient decline during the pandemic due to strict public health measures, and a subsequent resurgence after their relaxation, as GAS is transmitted via aerosols [10]. This rebound has been attributed not only to reduced population immunity, particularly among young children who had limited exposure to GAS during the pandemic years, but also to the return of seasonal respiratory viruses such as influenza and RSV, which are known to exacerbate iGAS infections. These factors together have likely amplified both the incidence and severity of iGAS in the post-pandemic period [10]. In children, iGAS infections are linked to peaks in respiratory syncytial virus (RSV) infections [9]. In late 2022, the WHO issued a warning about the resurgence of GAS infections in Europe. Several fatal pediatric cases were reported in a short time, mainly in the UK [3]. Belgium, where our hospital is located, was no exception. A recent article warned of a rise in iGAS cases in the Brussels-Capital Region [29]. This national trend was further confirmed by a genomic surveillance study conducted by the Belgian National Reference Center for invasive β-haemolytic streptococci, which reported a marked increase in emm1 iGAS bloodstream infections since mid-2022. This surge coincided with the emergence and rapid predominance of the toxigenic M1UK lineage of *S. pyogenes*, now accounting for the majority of emm1 bloodstream infections in Belgium [30]. Similarly, French pediatric surveillance data reported a marked post-pandemic rise in GAS infections, particularly among young children, underscoring the role of delayed immune exposure and viral co-circulation in this resurgence [2].

Within our hospital network, the Iris South Hospitals (a 550-bed public network in Brussels), we identified 14 pediatric cases of iGAS over the past 10 years, including 3 deaths and 4 with a documented link to viral infection (Table 2).

Three strains were typed emm6, one of which was involved in an osteoarticular infection. These local observations align with recent Belgian data reporting that over 50% of iGAS infections admitted to intensive care units were associated with septic shock, and that respiratory tract infections and viral co-infections, especially with influenza, were frequent predisposing factors [31]. Although macrolide and beta-lactam resistance among GAS strains is increasingly reported, our strain was susceptible to all antibiotics tested except quinolones [18]. This allowed effective treatment with an aminopenicillin followed by a switch to clindamycin, a lincosamide antibiotic related to macrolides, which has good bioavailability in hard-to-reach tissues such as deep abscesses, bones, and joints. Our microbiological investigation has two limitations. A throat swab could have revealed whether GAS colonization was the source of bacteremia in the context of mucosal damage from presumed viral pharyngitis. Also, pus from the abscess could not be obtained for culture due to the surgical risk to the patient.

## 4. Conclusions

To our knowledge, this is the first clearly documented case of a subperiosteal abscess and the first report of an emm6-type *S. pyogenes* strain involved in an invasive infection. Osteoarticular infections due to GAS have been documented in children, most often as septic arthritis or combined arthritis–osteomyelitis [32,33]. Additional pediatric cases include clavicular and even rib osteomyelitis [34,35]. In European pediatric cohorts, GAS is an uncommon but recognized pathogen among osteoarticular infections [36]. This background further underscores the exceptional nature of this subperiosteal long-bone presentation. This case highlights the importance of molecular typing in the epidemiological surveillance of invasive strains and the need for increased vigilance regarding unusual clinical presentations in children.

Despite ongoing efforts, there is currently no licensed vaccine against *S. pyogenes*. Several strategies are under development. The most advanced candidates are multivalent M-protein-based vaccines, such as the 30-valent StreptAnova, which notably includes emm6 among its targeted types. However, the wide diversity of circulating emm types worldwide continues to pose a challenge for universal protection. In parallel, non-M-protein-based approaches are being explored, targeting conserved virulence factors such as streptococcal exotoxins, surface enzymes, or adhesins, which could potentially provide broader strain coverage [37].

This case therefore underlines not only the clinical relevance of rare emm types in iGAS but also the importance of continued genomic surveillance and inclusive vaccine development strategies to address the full spectrum of circulating GAS strains.

## Figures and Tables

**Figure 1 microorganisms-13-02475-f001:**
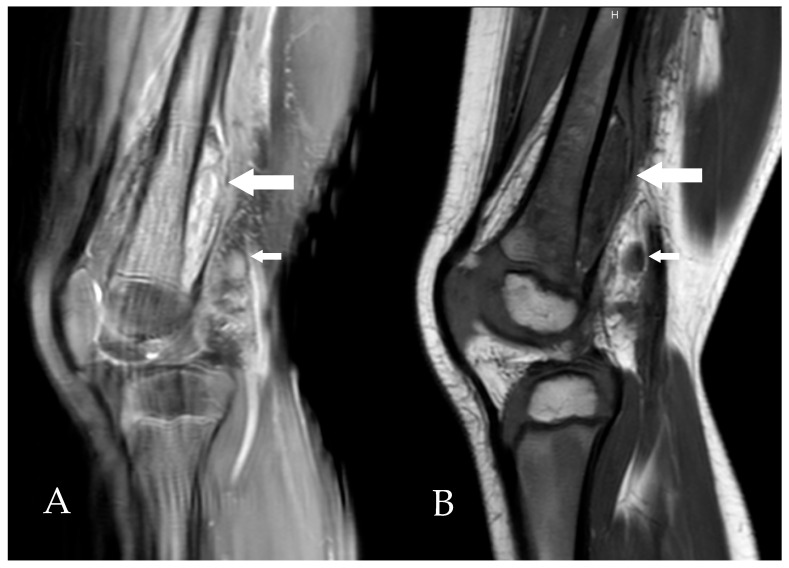
Sagittal MRI slices of the right knee, T2-weighted (**A**) and T1-weighted (**B**), showing a heterogeneous subperiosteal fluid collection (large arrows) and the presence of lymphadenopathy (small arrows).

**Figure 2 microorganisms-13-02475-f002:**
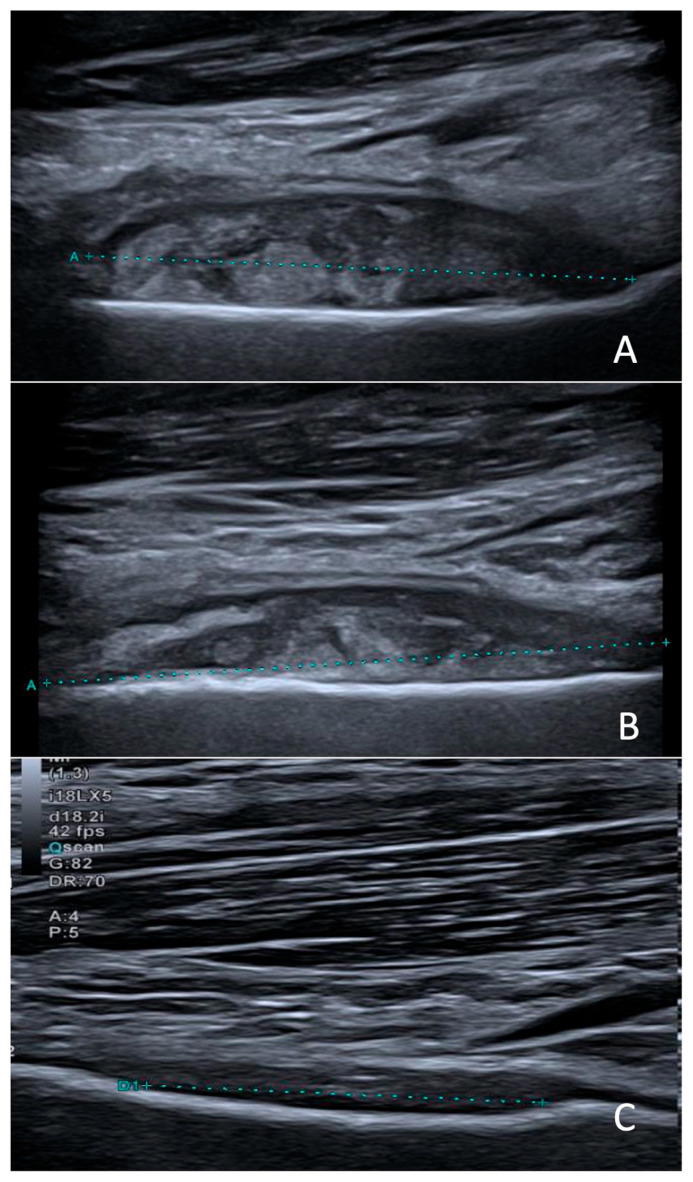
Sagittal ultrasound scans obtained at diagnosis (**A**), two weeks later (**B**), and three months after treatment initiation (**C**), showing progressive regression of the subperiosteal collection (between calipers).

**Table 1 microorganisms-13-02475-t001:** Virulence factors detected in the *S. pyogenes* emm6.10 isolate.

Virulence Factor Category	Virulence Genes	Virulence Factor
Adherence	*fbp54*, *lmb*, *cpa*, *sclA*, *sclB*, *plr/gapA*, *scpA/scpB*, *eno*	Fibronectin, laminin, collagen-binding proteins, C5a peptidase
Exoenzyme	*sda*, *mf/spd*, *mf2*, *mf3*, *speB*, *ska*, *cppA*, *htrA/degP*, *slaA*, *hylA*, *hylP*	Streptodornase, mitogenic factors, Streptokinase, proteases, hyaluronidase
Exotoxin	*speA*, *speC*, *speG*, *speK*, *smeZ*, *speH*, *slo*, *sagA*, *spyA*	Streptococcal exotoxins (Spe superantigens), hemolysins (Streptolysin O and Streptolysin S)
Immune modulation	*ideS/mac*, *endoS*, *hasA*, *hasB*, *hasC*, *galU*, *rfbB*	Immune evasion enzymes (IgG protease), hyaluronic acid capsule
Nutritional/Metabolic factor	*psaA*, *isdE*, *shr*	PsaA, NEAT-type surface protein direct heme uptake system
Stress survival	*tig/ropA*	Trigger factor involved in stress tolerance

**Table 2 microorganisms-13-02475-t002:** Pediatric iGAS Cases Recorded in Our Institution from January 2015 to December 2024.

Year	Sex	Age	Sample Type	Survival	emm Typing	Other Positive GAS Sites	Clinical Presentation and Associated Infections
2015	F	17 mo	Blood	No	12	–	Herpetic stomatitis; Purpura fulminans
2016	M	12 mo	Blood	Yes	6	–	–
2016	F	3 y	Blood	Yes	1	Pleural	Complicated pneumonia; Family viral illness
2017	F	8 mo	Blood	No	1	–	Cervical abscess
2018	M	15 mo	Blood	Yes	1	Throat	Pneumonia
2018	M	2 y	Blood	Yes	6	–	Ethmoiditis
2018	F	5 y	Blood	Yes	4	–	Septic osteomyelitis and elbow arthritis
2019	M	5 y	Blood	Yes	6	–	Cervical adenitis; Hip arthritis
2021	F	3 y	Blood	Yes	87	–	Pharyngitis; Enterovirus infection
2022	M	13 mo	Blood	Yes	75	Ear	Mastoiditis; Influenza A infection
2022	F	4 y	Blood	No	1	Pleural	Toxic shock syndrome; Varicella pneumonia
2022	F	17 mo	Blood	Yes	N/A	Ear	Bilateral perforated otitis media
2023	F	15 mo	Blood	Yes	1	–	Bacteremia without focus
2024	M	5 y	Blood	Yes	N/A	Ear	Rhinovirus infection

Abbreviations: GAS = group A *Streptococcus*; iGAS = invasive GAS infection; N/A = not available.

## Data Availability

The original contributions presented in this study are included in the article. Further inquiries can be directed to the corresponding author.

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
