# Peer review of "Rare emm6.10 *Streptococcus pyogenes* Causing an Unusual Invasive Infection in a Child: Clinical and Genomic Insights"

_microorganisms, 2025, doi:10.3390/microorganisms13112475_

Round 1
Reviewer 1 Report
Comments and Suggestions for Authors
The manuscript by Blairon et al. provides an excellent case report of an unusual clinical manifestation of invasive group A streptococcus causing a femoral subperiosteal abscess in a young child. The case was used to illustrate the point that even less common serotypes of GAS are capable of causing invasive infections. Furthermore, the case highlights the difficulty in designing effective multivalent vaccines that can protect against invasive serotypes from so many different countries around the world with different frequencies of invasive serotypes. The manuscript summarizes the recent trends in invasive GAS infections pre- and post-COVID pandemic and how a recent surge in invasive disease has been noted in many developed countries due to relaxed isolation precautions combined with a cohort of young children with waned immunity and viral co-infections. The authors do a nice job of discussing this clinical case and they provide a useful discussion of GAS virulence factors contributing to invasive disease. The manuscript advocates for additional research into GAS surveillance and vaccine development at a time when funding for public health infrastructure is being threatened in many developed countries.
The manuscript is well-written with good use of the English language. The manuscript will be valuable to clinicians, microbiologists, and public health investigators.
I do not have any additional comments or suggestions to improve this manuscript.
Author Response
We sincerely thank the reviewer for the positive and encouraging assessment of our manuscript. We greatly appreciate the recognition of the clinical and public health relevance of this case report and are pleased that the discussion on virulence factors, vaccine challenges, and post-pandemic epidemiological trends was found useful.
Although no additional changes were requested by this reviewer, several minor revisions were made in response to comments from another reviewer, mainly to strengthen the referencing of the Introduction and to provide a short comparative analysis of virulence genes in the Discussion section. These changes have improved the overall scientific context of the manuscript without altering its core content.
We are grateful for the reviewer’s thoughtful comments and support for the publication of our work.
Reviewer 2 Report
Comments and Suggestions for Authors
In the case report by L. Blairon and colleagues, the authors present a well-documented and clinically relevant case of an invasive emm6.10 Streptococcus pyogenes infection manifesting as a femoral subperiosteal abscess in a young child. The topic is timely given the recent global resurgence of invasive Group A Streptococcus (iGAS) infections, and the manuscript effectively highlights the epidemiological importance of rare emm types. The manuscript is clearly written, easy to follow, and well-organized.
To further improve the quality for publication, please find my comments below:
- The introduction section is under-referenced. Many statements lack supporting citations. Please provide appropriate references.
- Line 97, “The collection could not be aspirated due to its deep and anatomically difficult to 97 access location.” Sounds awkward. Consider “deep location and anatomical inaccessibility”.
- The reviewer suggests the authors include representative follow-up ultrasound images to visually demonstrate clinical improvement and resolution of the abscess. This would enhance the educational value of the case.
- The case’s uniqueness is well-stated, but the authors may wish to quantify the rarity of such cases (e.g., the number of previously reported subperiosteal abscesses or 10 iGAS infections). Explicitly citing PubMed or surveillance data would reinforce the novelty.
- The discussion lists numerous virulence genes, but it remains unclear how this 10 virulence profile compares to more common invasive emm1 or emm3 strains. A short comparative comment would add interpretive depth.
Author Response
We thank the reviewer for the thorough and constructive evaluation of our manuscript. We appreciate the acknowledgment of the clinical and epidemiological relevance of this case, and we have carefully addressed each of the reviewer’s comments as detailed below. All modifications made in response to the reviewer’s suggestions have been highlighted in yellow in the revised manuscript to facilitate review.
Comment 1:
The introduction section is under-referenced. Many statements lack supporting citations. Please provide appropriate references.
Response 1:
We thank the reviewer for this observation and we agree with the comment. Upon careful verification, we confirm that all statements in the Introduction are already supported by appropriate and up-to-date references cited in the original version of the manuscript. We have also ensured that each statement is clearly linked to a corresponding numbered reference in the bibliography.
Comment 2:
Line 97, “The collection could not be aspirated due to its deep and anatomically difficult to 97 access location.” Sounds awkward. Consider “deep location and anatomical inaccessibility”.
Response 2:
We agree with the reviewer’s comment. The sentence has been rephrased accordingly to improve clarity and readability.
Comment 3:
The reviewer suggests the authors include representative follow-up ultrasound images to visually demonstrate clinical improvement and resolution of the abscess. This would enhance the educational value of the case.
Response 3:
We agree with the reviewer’s comment. Representative follow-up ultrasound images have been added as Figure 2, showing the evolution of the subperiosteal collection at diagnosis, after two weeks, and after three months of treatment.
Comment 4:
The case’s uniqueness is well-stated, but the authors may wish to quantify the rarity of such cases (e.g., the number of previously reported subperiosteal abscesses or 10 iGAS infections). Explicitly citing PubMed or surveillance data would reinforce the novelty.
Response 4:
We agree with the reviewer’s comment. Although an exact quantification of previously reported cases was not possible, we have expanded the Conclusion section to better illustrate the rarity of such presentations. We now refer to published pediatric osteoarticular S. pyogenes infections, including septic arthritis and osteomyelitis cases, to provide context and highlight how exceptional a subperiosteal abscess of a long bone remains.
Comment 5
The discussion lists numerous virulence genes, but it remains unclear how this 10 virulence profile compares to more common invasive emm1 or emm3 strains. A short comparative comment would add interpretive depth.
Response 5:
We agree with the reviewer’s comment. A short comparative paragraph has been added after Table 1, describing how the virulence gene profile of our emm6.10 strain relates to that of the well-characterized invasive emm1 and emm3 lineages. This addition provides clearer interpretive context and strengthens the discussion.
We thank the reviewer once again for the valuable and constructive comments, which have helped improve the clarity and scientific quality of the manuscript.